# Systematic analysis of non-melanoma skin cancer burden: A comparative study between China and the world from 1990 to 2021 and prediction to 2036

Su Liang[1], Tao Sun[2], Xuesong Jia[1]*, Juanmei Cao [1]*, Xue Wang[1]*

1 Department of Dermatology, the First Affiliated Hospital of Shihezi University, Shihezi, Xinjiang, China,
2 STD & AIDS Prevention and Control Division, Shihezi Center for Disease Control and Prevention, Shihezi, Xinjiang, China

☯ Contributed equally to this work as co-first authors.
* jiaxs0309@sina.com (XJ); 124739471@qq.com (JC); 1658436810@qq.com (XW)

## Abstract

### Objective

To compare the characteristics and trends of the non-melanoma skin cancer (NMSC) burden in China and globally, and to provide a basis for the development of effective prevention and control measures in China.

### Methods

Data related to the incidence, death, and disability-adjusted life years (DALYs) of NMSC in China and the world were obtained from the Global Burden of Disease (GBD) 2021 database. The population included in this sutdy were adults aged 20 and above. The Joinpoint regression model was used to estimate the average annual percentage of change (AAPC) to reflect the time trend. Bayesian age-period-cohort model was constructed for prediction, and the coefficient of determination $R^2$ was used to reflect the fitting degree of the model.

### Results

From 1990 to 2021, the age-standardized incidence rate, mortality rate and DALYs rate of NMSC in China increased by 707.31%, 16.00% and 10.04%, respectively, all higher than the global level. And the corresponding upward trends were also more obvious, with AAPC of 6.71% (95%: 6.01% − 7.18%), 0.46% (95%: 0.40% − 0.52%) and 0.28% (95%: 0.22% − 0.34%), respectively. The incidence rate and its AAPC of NMSC were higher in men than in women. Moreover, the mortality rate and DALYs rate were also higher in men, but the growth rate was faster in women. The high incidence rate, mortality rate, and DALYs rate of NMSC all occurred in the higher

**Data availability statement:** All data can be publicly obtained from link https://www.healthdata.org/Data-tools-practices/data-practices/ihme-free-charge-non-commercial-user-agreement.

**Funding:** This study was supported by the Research Project of National Natural Science Foundation of China in the form of a grant awarded to X.W (Grant No: 82460624), the Guiding Plan Project of Shihezi Science and Technology Bureau in the form of a grant awarded to S.L (Grant No: 2024ZDYL07), the Science and Technology Program of XPCC in the form of a grant awarded to JM.C (Grant No: 2023ZD023), the Science and Technology Program of XPCC in the form of a grant awarded to XS.J (Grant No: 2024ZD042), and the Science and Technology Program of XPCC in the form of a grant awarded to X.W (Grant No: 2025DA051). The specific roles of this author are articulated in the 'author contributions' section. The funders had no role in study design, data collection and analysis, decision to publish, or preparation of the manuscript.

**Competing interests:** The authors have declared that no competing interests exist.

age groups. The age-standardized incidence rates of NMSC in China and globally were predicted to continue to rise over the next 15 years, while the age-standardized mortality rates will decline.

## Conclusion

The burden of NMSC in China remained serious, especially in the context of an increasingly aging population. Relevant authorities should continue to develop and optimize preventive and control measures, especially for men, and adopt targeted measures to reduce the burden of NMSC significantly.

## 1. Introduction

Non-melanoma skin cancer (NMSC) was one of the most common cancers, divided into basal cell carcinoma and squamous cell carcinoma, and had become a prominent public health problem worldwide [1]. According to GLOBOCAN, there were 1,234,595 new cases and 69,481 deaths of NMSC globally in 2022, accounting for 0.7% of all cancer deaths [2]. The incidence of NMSC had been found to increase annually, especially in areas with high ultraviolet (UV) exposure, such as Oceania and parts of the United States, where the incidence was highest [3], and was more common in the white population [4]. The age-standardized incidence rate (ASIR) of NMSC in the United States had increased from 402 (/100,000) in 1990787 (/100,000) in 2019 [5]. However, the actual incidence of NMSC may be underestimated, potentially due to the following reasons: many countries did not require reporting NMSC information to national cancer registries, which directly led to most countries not giving sufficient attention to NMSC [6–9]; the high incidence of NMSC not only resulted in a large workload and difficulty in implementing comprehensive registration, but also posed technical challenges in accurately recording each patient's tumor status and extracting structured pathological data, making it difficult to obtain relevant data [10]. These factors collectively contribute to the fact that the actual burden of NMSC remains unclear and is frequently underestimated [11].

Compared with melanoma, NMSC has a lower mortality rate and better prognosis. This difference has led to limited scientific evidence on the epidemiological characteristics and burden of NMSC in China [12]. Epidemiological evidence was essential for effective disease prevention and control. Therefore, in order to better address the challenges posed by NMSC, this study, based on the Global Burden of Disease (GBD) 2021 database, used the Joinpoint regression model and the Bayesian age-period-cohort (BAPC) model first to analyze and predict the burden (incidence rate, mortality rate, and disability-adjusted life years (DALYs)) caused by NMSC in China from 1990 to 2021 by gender, and then compared it with the global situation, to provide a basis for formulating more effective prevention and treatment strategies and intervention measures.

## 2. Materials and methods

### 2.1 Data collection

Data related to the burden of NMSC were obtained from the GBD 2021 database, which provided epidemiological data for 204 countries or regions, 371 diseases, and 88 risk factors, and provided a strong basis for detailed and extensive understanding of global health trends and emerging challenges [13]. This study was based on the GBD 2021 database and used the C44 code in the International Classification of Diseases, Tenth Revision (ICD-10) to define NMSC [13]. This database provided the number of new cases, deaths, disability-adjusted life years (DALYs) and their corresponding rates of NMSC for population aged 20 and above, categorized by different genders [13].

### 2.2 Data analysis

Using the GBD 2021 world standard population structure, this study applied the direct method to age-standardise the incidence, mortality, and DALYs rates of NMSC in China and globally. This method was based on the assumption that these indicators followed an independent Poisson random variable weighted sum distribution. By controlling for changes in population age structure, it accurately quantified differences in the burden of NMSC across different time periods, genders, and regions. The calculation formula was as follows:

$$Age-standardized\ rate = \frac{\sum_{i=1}^{A} a_i W_i}{\sum_{i=1}^{A} W_i}$$

Where $a_i$ indicates the age-specific rate for the i-th age group, $W_i$ indicates the weight of the corresponding age subgroup in the selected reference population (i represents the i-th age category), and A indicates the total number of age groups.

#### 2.2.1 Joinpoint regression model.
The Joinpoint regression model was a log-linear model with time (year) as the independent variable and incidence or mortality as the dependent variable [14]. It described the trend of incidence or mortality by splicing the joinpoints of different logarithmic line segments [15]. This model established segmented regression based on the temporal characteristics of disease distribution, dividing the study period into different intervals using multiple connection points, and fitting and optimizing the trends within each interval to assess the characteristics of specific disease changes within different intervals [14]. During model construction, Monte Carlo simulation tests were used to determine the number of connection points, their positions, and corresponding P-values. A two-sided test was employed with a significance level of 0.05, and the Bonferroni method was applied to correct the significance level and control the false positive rate [14]. In this study, the number of joint points was set within the range of 0–6 to ensure the model adequately captures temporal trend changes while avoiding overfitting. The model was used to calculate the average annual percentage change (AAPC) and its 95% confidence interval (CI) to reflect the trend over time. The model formula was as follows [14]:

$$E[y \mid x] = \beta_0 + \beta_1 x + \delta_1 (x - \tau_1)^+ + ... + \delta_k (x - \tau_k)^+$$

When both AAPC and its 95% CI are > 0, it meant that the corresponding indicator was on the rise; both < 0 were on the decline; and containing 0 meant that the indicator remained stable.

#### 2.2.2 Bayesian age-period-cohort model.
We used the BAPC model to predict the age-standardized incidence, mortality, and DALYs rate for NMSC by sex from 2022 to 2036 [16]. The model allowed for the inference of unknown parameters by combining the posterior information obtained from the sample information with the a priori information on the unknown parameters [17], a process implemented using the BAPC and Integrated Nested Laplace Approximation (INLA) packages in R 4.3.2. To verify the accuracy of the model, we calculated the coefficient of determination (R²) for

each BAPC model. R² reflects the extent to which the model explains the variation in the data. The closer the value is to 1, the better the model fit and the higher the prediction accuracy.

## 2.3 Statistic analysis

This study used the Joinpoint Regression Program (version 5.0.2) developed by the U.S. National Cancer Institute to perform Joinpoint regression analysis. Graphical visualization was conducted using R software (version 4.3.2, https://www.r-project.org/), and the BAPC and INLA packages were utilized to construct the BAPC model.

## 3. Results

### 3.1 Incidence of NMSC in China and the world

Compared with 1990, the number of global NMSC cases in 2021 increased from 1,661,600–6,336,800, with an increased rate of 281.37% and an upward trend for AAPC = 4.37% (95%CI: 4.29% ~ 4.44%); the incidence rate and ASIR increased from 31.15/100,000 and 45.04/100,000 to 80.30/100,000 and 74.10/100,000, respectively, with an increased rate of 157.78% and 64.52%, showed an upward trend, and the AAPC was 3.06% (95%CI: 2.93% ~ 3.15%) and 1.57% (95%CI: 1.49% ~ 1.64%), respectively (Table 1).

The number of NMSC cases in China increased from 39,500–791,900, and the increased rate (1904.81%) and upward trend (AAPC = 9.88% (95%CI: 9.16% ~ 10.37%)) were higher than that of the global level; the incidence rate and ASIR increased from 3.36/100,000 and 4.65/100,000 in 1990 to 55.66/100,000 and 37.54/100,000 in 2021, respectively, and both were lower than the global level in the same period, but their increase rates (1556.55% and 707.31%) and AAPC (9.22% (95%CI: 8.51% ~ 9.71%) and 6.71% (95%CI: 6.01% ~ 7.18%)) were significantly higher than the global level (Table 1).

**Table 1. The incidence of NMSC in China and global, 1990-2021.**

| Sex | Incidence cases (10,000 cases) | | Incidence rate (per 100,000) | | ASIR (per 100,000) | |
|---|---|---|---|---|---|---|
| | China | Global | China | Global | China | Global |
| Both | | | | | | |
| 1990 | 3.95 | 166.16 | 3.36 | 31.15 | 4.65 | 45.04 |
| 2021 | 79.19 | 633.68 | 55.66 | 80.30 | 37.54 | 74.10 |
| Change (%) | 1904.81 | 281.37 | 1556.55 | 157.78 | 707.31 | 64.52 |
| AAPC (95%CI) (%) | 9.88 (9.16 ~ 10.37) | 4.37 (4.29 ~ 4.44) | 9.22 (8.51 ~ 9.71) | 3.06 (2.93 ~ 3.15) | 6.71 (6.01 ~ 7.18) | 1.57 (1.49 ~ 1.64) |
| Male | | | | | | |
| 1990 | 2.15 | 87.40 | 3.55 | 32.54 | 5.34 | 55.30 |
| 2021 | 44.37 | 369.63 | 60.94 | 93.36 | 43.66 | 95.82 |
| Change (%) | 1963.72 | 322.92 | 1616.62 | 186.91 | 717.60 | 73.27 |
| AAPC (95%CI) (%) | 9.99 (9.29 ~ 10.47) | 4.72 (4.63 ~ 4.79) | 9.36 (8.67 ~ 9.84) | 3.44 (3.31 ~ 3.53) | 6.77 (6.10 ~ 7.22) | 1.79 (1.69 ~ 1.88) |
| Female | | | | | | |
| 1990 | 1.80 | 78.77 | 3.15 | 29.75 | 4.11 | 38.23 |
| 2021 | 34.82 | 264.05 | 50.12 | 67.16 | 32.57 | 57.33 |
| Change (%) | 1834.44 | 235.22 | 1491.11 | 125.75 | 692.46 | 49.96 |
| AAPC (95%CI) (%) | 9.75 (8.99 ~ 10.25) | 3.87 (3.76 ~ 3.96) | 9.06 (8.31 ~ 9.56) | 2.63 (2.47 ~ 2.73) | 6.62 (5.87 ~ 7.12) | 1.20 (1.09 ~ 1.29) |

NMSC, Non-melanoma skin cancer; ASIR, Age-standardized incidence rate; AAPC, Average annual percentage of change.

Both in China and the world, the number of cases, incidence rate, ASIR, as well as the corresponding increase rate and AAPC of NMSC were all higher in men than in women. In China, the age group with the highest number of NMSC cases was 65–74 years old, and the lowest was over 85 years old. Globally, the age group with the highest number of cases was 65–74 years old, and the lowest was 55–59 years old. The age group with the lowest incidence rate in both China and globally was 20–54 years old. The age group with the highest incidence rate in China was 80–84 years old, while globally it was those above 85 years old (Table 1, Fig 1).

We used the BAPC model to predict the ASIR of NMSC in China and globally based on different genders. The model fit was excellent ($R^2_{China\ male} = 0.99998$, $R^2_{China\ female} = 0.99998$; $R^2_{Global\ male} = 0.96204$, $R^2_{Global\ female} = 0.98055$). In 2022–2036, the ASIR of the Chinese population showed an increasing trend, with the male and female ASIR increasing from 79.94/100,000 and 63.35/100,000 in 2022–984,734.59/100,000 and 1,007,157.09/100,000 in 2036, respectively; and the global ASIR showed an increasing trend, with the male and female ASIR increased from 97.89/100,000 and 59.13/100,000 in 2022 to 173.16/100,000 and 111.51/100,000 in 2036 (Fig 2).

## 3.2 Mortality of NMSC in China and the world

Compared with 1990, the global deaths in 2021 increased from 22,700–56,900, with an increase of 150.66% and an upward trend for AAPC = 3.03% (95%CI: 3.00%~3.05%); the mortality rate and the age-standardized mortality rate

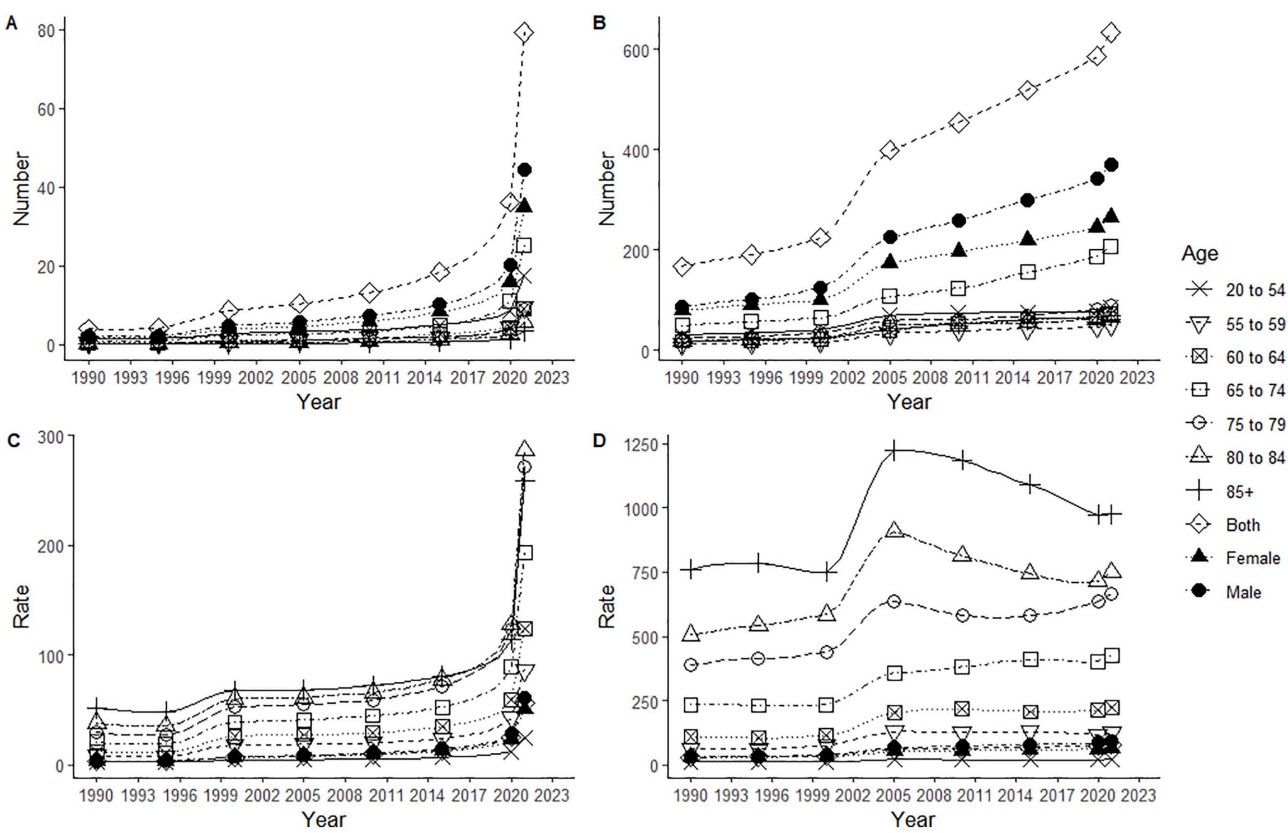

**Fig 1. Incidence changes of NMSC in China and global from 1990 to 2021.** Incidence cases in China (A) and Global (B), incidence in China (C) and Global (D).

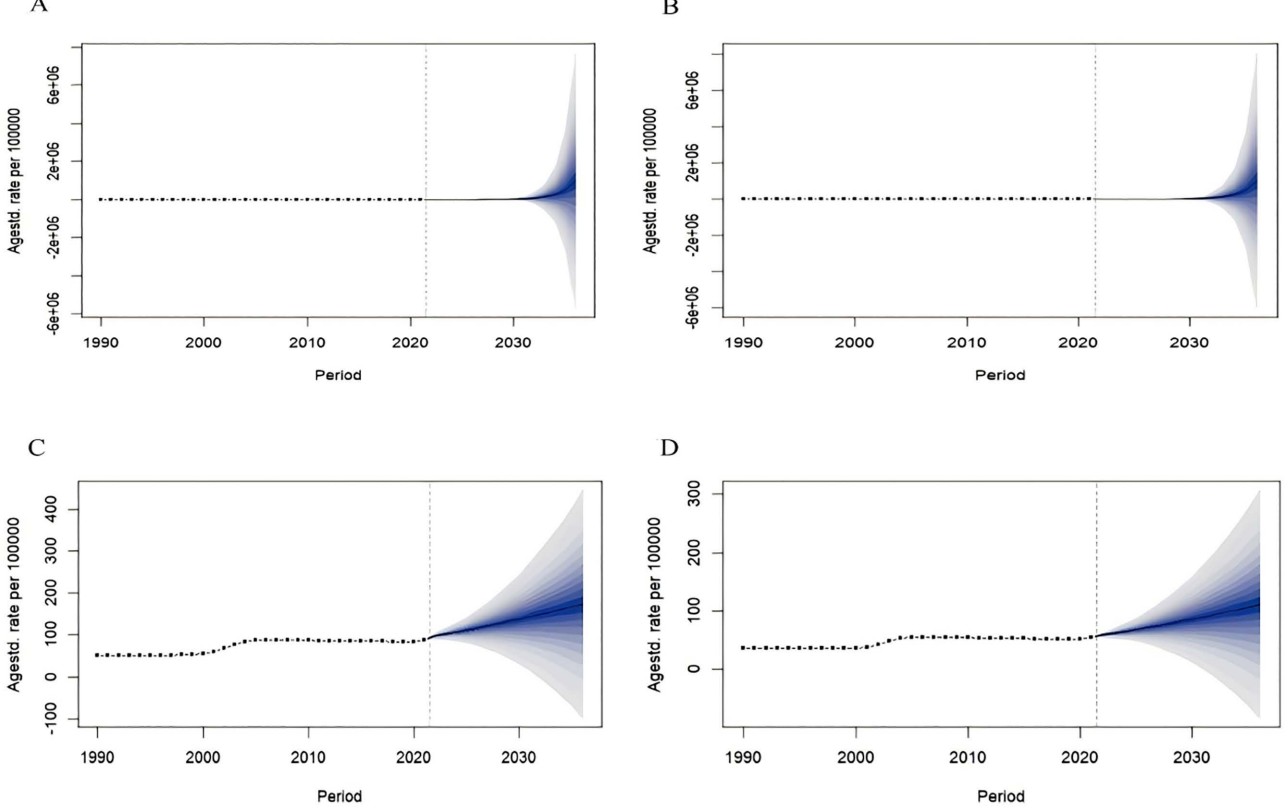

**Fig 2. The ASIR prediction of NMSC in China and global from 2022 to 2036.** ASIR prediction in China male (A), China female (B), Global male (C), Global female (D). Observed (dashed lines) and predicted rates (solid lines). The blue region shows the upper and lower limits of the 95% uncertainty intervals (95% UI). ASIR, Age-standardized incidence rate. NMSC, Non-melanoma skin cancer.

(ASMR) increased from 0.42/100,000 and 0.67/100,000 to 0.72/100,000 and 0.69/100,000, respectively, with an increased rate of 71.43% and 2.99%, and an AAPC of 1.74% (95%CI: 1.71%~1.76%) and 0.14% (95%CI: 0.12%~0.16%), respectively (Table 2).

The NMSC deaths in China increased from 5,200–16,600, and the increased rate (219.23%) and upward trend (AAPC = 3.75% (95%CI: 3.68%~3.80%)) were higher than that of the global level; the mortality rate and ASMR increased from 0.44/100,000 and 0.75/100,000 in 1990 to 1.17/100,000 and 0.87/100,000 in 2021, respectively, which were consistently higher than the global level during the same period, as were the increased rates (165.91% and 16.00%) and AAPC (3.17% (95%CI: 3.09%~3.22%) and 0.46% (95%CI: 0.40%~0.52%) (Table 2).

Both in China and the world, the deaths, mortality rates and ASMR for NMSC were higher in men than in women, and the corresponding increase rates and AAPC were higher in men globally, but lower in men in China. The age group with the highest NMSC deaths in China was 65–74 years, and the lowest was 55–59 years; the age group with the highest deaths globally was 85 + years, and the lowest was 55–59 years. The lowest age group for mortality in both China and globally was 20–54 years, and the highest was 85 + years (Table 2, Fig 3).

We used the BAPC model to predict the ASMR of NMSC in China and globally based on different genders. The model fit was excellent ($R^2_{China\ male}$ = 0.99643, $R^2_{China\ female}$ = 0.98751; $R^2_{Global\ male}$ = 0.86609, $R^2_{Global\ female}$ = 0.98699). Both the Chinese and global population-based ASMR were predicted to show a decreasing trend in 2022–2036, with the Chinese male and female ASMR decreased from 0.88/100,000 and 0.68/100,000 in 2022 to 0.82/100,000 and 0.56/100,000 in

**Table 2. The mortality of NMSC in China and global, 1990-2021.**

| Sex | Deaths (10,000 cases) | | Mortality rate (per 100,000) | | ASMR (per 100,000) | |
|---|---|---|---|---|---|---|
| | China | Global | China | Global | China | Global |
| Both | | | | | | |
| 1990 | 0.52 | 2.27 | 0.44 | 0.42 | 0.75 | 0.67 |
| 2021 | 1.66 | 5.69 | 1.17 | 0.72 | 0.87 | 0.69 |
| Change (%) | 219.23 | 150.66 | 165.91 | 71.43 | 16.00 | 2.99 |
| AAPC (95%CI) (%) | 3.75 (3.68~3.80) | 3.03 (3.00~3.05) | 3.17 (3.09~3.22) | 1.74 (1.71~1.76) | 0.46 (0.40~0.52) | 0.14 (0.12~0.16) |
| Male | | | | | | |
| 1990 | 0.28 | 1.28 | 0.46 | 0.48 | 0.93 | 0.88 |
| 2021 | 0.85 | 3.22 | 1.17 | 0.81 | 1.03 | 0.92 |
| Change (%) | 203.57 | 151.56 | 154.35 | 68.75 | 10.75 | 4.55 |
| AAPC (95%CI) (%) | 3.59 (3.51~3.67) | 3.05 (3.03~3.07) | 2.94 (2.86~3.01) | 1.75 (1.73~1.77) | 0.23 (0.09~0.33) | 0.16 (0.14~0.17) |
| Female | | | | | | |
| 1990 | 0.24 | 0.99 | 0.42 | 0.37 | 0.64 | 0.52 |
| 2021 | 0.81 | 2.47 | 1.16 | 0.63 | 0.77 | 0.53 |
| Change (%) | 237.50 | 149.49 | 176.19 | 70.27 | 20.31 | 1.92 |
| AAPC (95%CI) (%) | 3.97 (3.91~4.02) | 3.03 (2.98~3.07) | 3.38 (3.30~3.46) | 1.72 (1.68~1.75) | 0.60 (0.53~0.67) | 0.08 (0.04~0.11) |

NMSC, Non-melanoma skin cancer; ASMR, Age-standardized mortality rate; AAPC, Average annual percentage of change.

2036, respectively, and the global male and female ASMR decreased from 0.78/100,000 and 0.45/100,000 in 2022 to 0.67/100,000 and 0.39/100,000 in 2036 (Fig 4).

### 3.3 DALYs of NMSC in China and the world

Compared to 1990, global DALYs in 2021 increased from 545,600–1,122,900 person-years, the increased rate was 122.31%, and the AAPC was 2.61% (95%CI: 2.59% to 2.63%); the DALYs rate and the age-standardized DALYs rate (ASDR) increased from 10.23/100,000 and 14.02/100,000 to 15.37/100,000 and 14.33/100,000, respectively, the increased rate was 50.24% and 2.21%, respectively, and the AAPC was 1.31% (95%CI: 1.29%~1.33%) and 0.07% (95%CI: 0.04%~0.08%), respectively (Table 3).

DALYs due to NMSC in China increased from 140,100–360,800 person-years, and the increased rate (151.53%) and AAPC (3.07% (95%CI: 2.97%~3.12%)) were higher than that of the global level; the DALYs rate and ASDR increased from 11.91/100,000 and 16.34/100,000 in 1990 to 25.36/100,000 and 17.98/100,000 in 2021, respectively, both of which were higher than the global level in the same period, and their increased rates (112.93% and 10.04%) and AAPC (2.45% (95%CI: 2.37%~2.50%) and 0.28% (95%CI: 0.22%~0.34%)) were also higher than the global level (Table 3).

Both in China and globally, DALYs, DALYs rate and ASDR were higher in men than in women, and the corresponding increased rate and AAPC were lower in Chinese men than in women. DALYs increased with age in both China and globally; the lowest DALYs rate in China was in the 20–54 age group and the highest in the 85+ age group; the lowest DALYs rate globally was in the 20–54 age group and the highest in the 85+ age group (Table 3, Fig 5).

We used the BAPC model to predict the ASDR of NMSC in China and globally based on different genders. The model fit was excellent ($R^2_{China\ male}$ = 0.99987, $R^2_{China\ female}$ = 0.99949; $R^2_{Global\ male}$ = 0.97468, $R^2_{Global\ female}$ = 0.99812). In 2022–2036, it was predicted that the ASDR in Chinese males will increase from 19.22/100,000 in 2022 to 21.79/100,000 in 2036; the

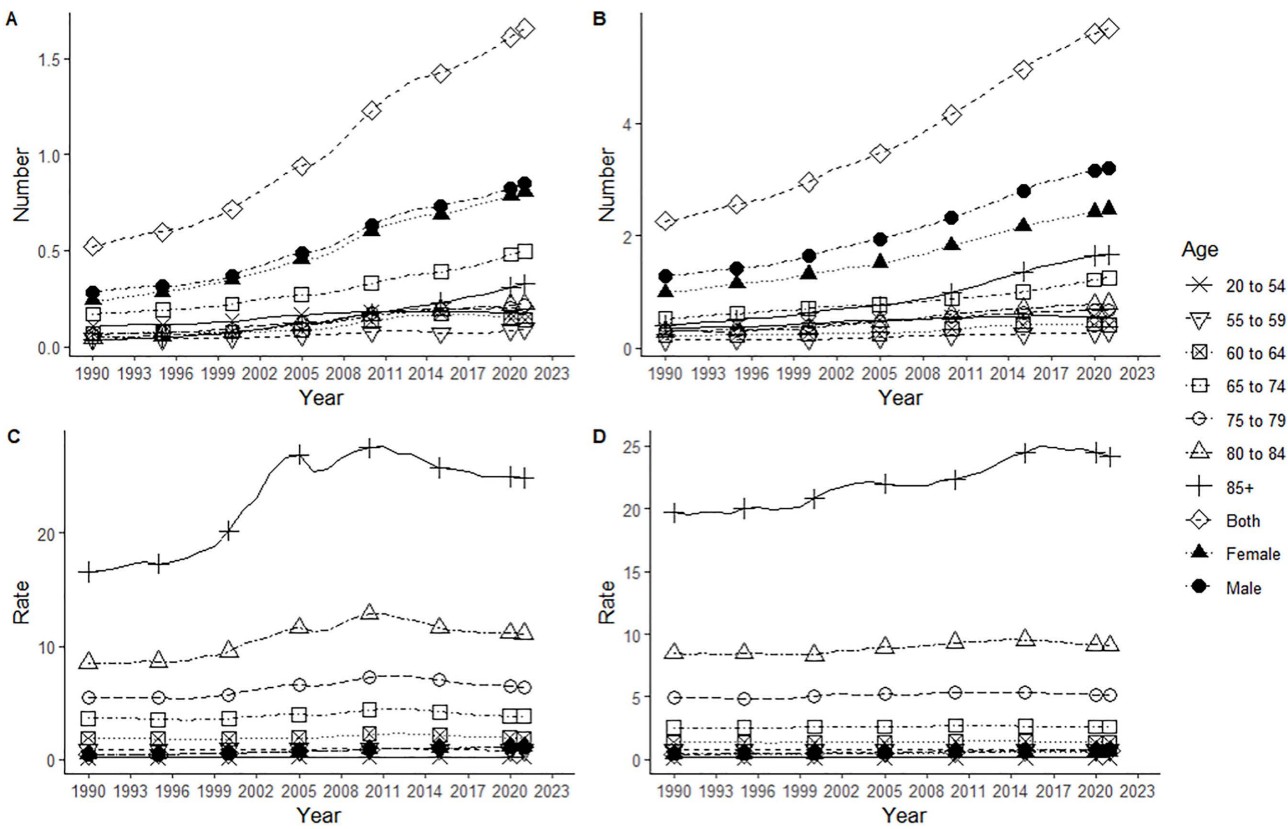

**Fig 3. Mortality changes of NMSC in China and global from 1990 to 2021.** Deaths in China (A) and Global (B), mortality rate in China (C) and Global (D).

ASDR in Chinese females will decrease from 15.44/100,000 in 2022 to 13.98/100,000 in 2036. The global ASDR in males and females will decrease from 17.24/100,000 and 10.13/100,000 in 2022 to 15.40/100,000 and 8.95/100,000 in 2036, respectively (Fig 6).

## 4. Discussion

In this study, the GBD 2021 database was used to comprehensively compare and analyze the incidence, mortality, and burden of NMSC in China and globally from 1990 to 2021, and to predict the trend in the next 15 years, providing strong evidence for the prevention and treatment of NMSC. Although the ASIR of NMSC in China was lower than the global level, the levels of both ASMR and ASDR were relatively higher. Meanwhile, the increased rates of the three indicators for NMSC in China were faster than those in the world. There were significant gender differences in the burden of NMSC: the incidence and its increase rate among men were both higher than those of women; the mortality and DALYs rate for men were more severe, but their increase rates for women were faster. The high incidence, mortality, and DALYs rate of NMSC occurred in the higher age groups. It is predicted that the ASIR of China and global NMSC will continue to rise in the next 15 years, while the ASMR will decline.

The burden of NMSC increased in China and globally from 1990 to 2021, and the increase rate in China was higher than the global level. This finding was consistent with another study, which indicated that between 1992 and 2021, China showed an upward trend in NMSC mortality rate, which remained higher than the global average [18]. This suggested that

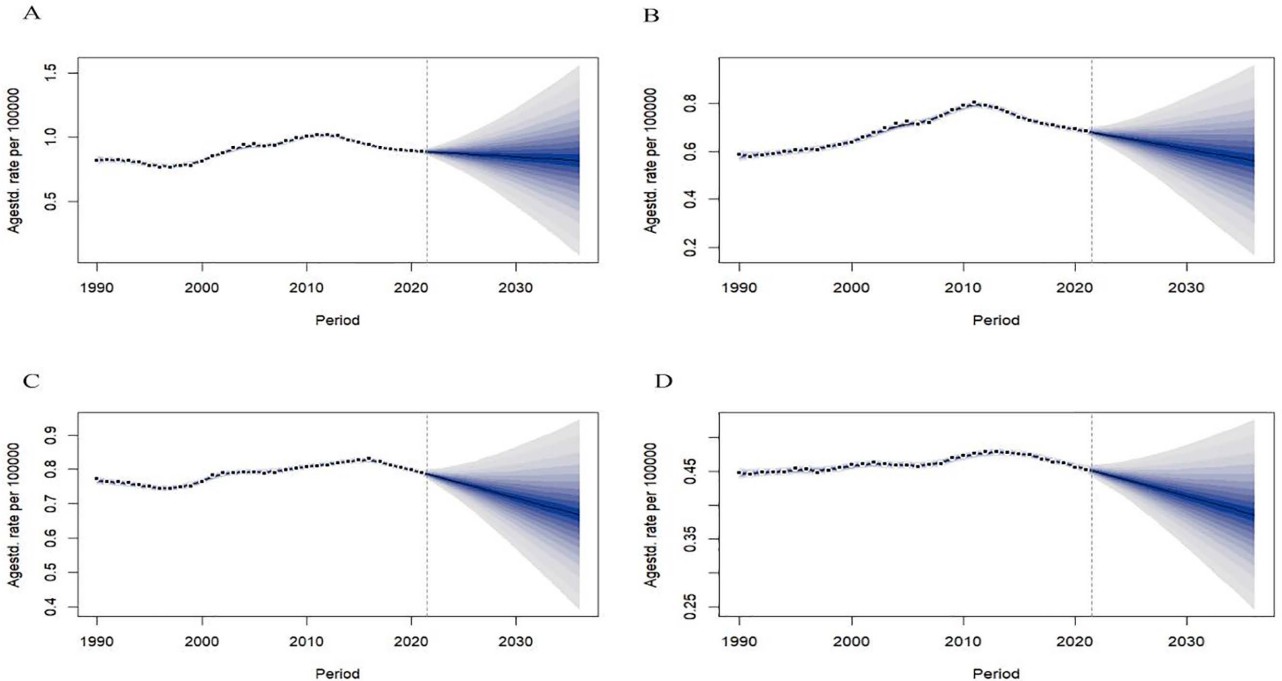

**Fig 4. The ASMR prediction of NMSC in China and global from 2022 to 2036.** ASMR prediction in China male (A), China female (B), Global male (C), Global female (D). Observed (dashed lines) and predicted rates (solid lines). The blue region shows the upper and lower limits of the 95% uncertainty intervals (95% UI). ASMR, Age-standardized mortality rate. NMSC, Non-melanoma skin cancer.

the NMSC continued to be a significant public health concern that cannot be overlooked. This may be influenced by multiple factors, including population aging, environmental pollution, ozone layer depletion, and climate change. As a high-risk group for NMSC, the global trend of an increasing proportion of elderly people had significantly impacted the incidence and mortality rates of this disease [19]. Additionally, UV radiation, a known risk factor for NMSC, played a key role in the pathogenesis of skin cancer by causing DNA damage and immunosuppression [5]. Climate change had caused various health issues worldwide, from heat-related illnesses to changes in the geographical distribution of infectious diseases and increased risks of zoonotic disease transmission, with its impact on human health becoming increasingly widespread [20]. Among these climate change-related health threats, the incidence of NMSC increased significantly, becoming one of the key manifestations of climate change's impact on human health [21]. Climate change accelerated ozone layer depletion, increasing surface UV radiation intensity and raising human exposure to UV radiation [22]. Additionally, socio-economic development and lifestyle changes exacerbated this issue. As economic levels improved, people had more opportunities to participate in outdoor activities and vacation travel, increasing the likelihood of sunburn. Sunburn was significantly associated with an increased risk of NMSC, particularly among women under 25. These factors intertwine, collectively contributing to the ongoing global rise in the burden of NMSC [23]. It is worth noting that during this period, compared to the global level, the incidence of NMSC in China was relatively low but increasing rapidly, while the mortality and DALYs were relatively high. This may be related to multiple factors. For instance, late diagnosis and uneven distribution of medical resources could be important reasons. Studies have shown that NMSC is often overlooked, leading to delayed diagnosis [11]. Underreporting of non-fatal cases is also a possible factor. Compared to other cancers with high mortality rates (such as lung cancer, gastric cancer, or pancreatic cancer), NMSC had a relatively low mortality rate. This may lead to healthcare resources and public health attention being directed more toward those more lethal cancer types. Due to

**Table 3. The DALYs of NMSC in China and global, 1990-2021.**

| Sex | DALYs (per 100,000) | | DALYs rate (per 100,000) | | ASDR (per 100,000) | |
|---|---|---|---|---|---|---|
| | China | Global | China | Global | China | Global |
| Both | | | | | | |
| 1990 | 14.01 | 54.56 | 11.91 | 10.23 | 16.34 | 14.02 |
| 2021 | 36.08 | 121.29 | 25.36 | 15.37 | 17.98 | 14.33 |
| Change (%) | 151.53 | 122.31 | 112.93 | 50.24 | 10.04 | 2.21 |
| AAPC (95%CI) (%) | 3.07 (2.97~3.12) | 2.61 (2.59~2.63) | 2.45 (2.37~2.50) | 1.31 (1.29~1.33) | 0.28 (0.22~0.34) | 0.07 (0.04~0.08) |
| Male | | | | | | |
| 1990 | 7.49 | 32.35 | 12.34 | 12.04 | 18.71 | 18.26 |
| 2021 | 18.85 | 71.78 | 25.88 | 18.13 | 20.23 | 18.61 |
| Change (%) | 151.67 | 121.89 | 109.72 | 49.75 | 8.12 | 1.92 |
| AAPC (95%CI) (%) | 3.01 (2.91~3.08) | 2.61 (2.59~2.63) | 2.36 (2.24~2.43) | 1.33 (1.30~1.35) | 0.18 (0.06~0.26) | 0.04 (0.01~0.07) |
| Female | | | | | | |
| 1990 | 6.52 | 22.21 | 11.45 | 8.39 | 14.77 | 10.63 |
| 2021 | 17.23 | 49.51 | 24.82 | 12.59 | 16.34 | 10.82 |
| Change (%) | 164.26 | 122.92 | 116.77 | 50.06 | 10.63 | 1.79 |
| AAPC (95%CI) (%) | 3.16 (3.10~3.20) | 2.63 (2.61~2.66) | 2.56 (2.48~2.62) | 1.31 (1.28~1.34) | 0.35 (0.29~0.41) | 0.07 (0.04~0.09) |

NMSC, Non-melanoma skin cancer; ASDR, Age-standardized disability-adjusted life years rate; AAPC, Average annual percentage of change; DALYs, Disability-adjusted life years.

the relatively low baseline incidence of NMSC in China, the healthcare system did not place sufficient emphasis on this disease, resulting in incomplete reporting of non-fatal NMSC cases [24].

Based on this trend, it is recommended to strengthen the early screening and diagnosis system for NMSC, especially in areas with scarce medical resources; conduct health education and regular check-ups for key populations such as the elderly and young women; incorporate NMSC prevention and control into climate change response strategies and strengthen the construction of UV radiation warning systems; optimize the allocation of medical resources to enhance primary healthcare facilities' diagnostic and treatment capabilities; more national and subnational epidemiological studies on NMSC should be conducted to establish prevention and control strategies and clinical guidelines tailored to the national context, providing scientific evidence for public health decision-making.

As one of the world's fastest-aging countries, China faced a particularly severe challenge [25]. In addition, since the reform and opening up, China's economy had maintained high growth while facing enormous environmental cost pressures. The dependence of China's energy consumption structure on fossil energy sources, such as coal, had led to increasing problems with pollutants and carbon emissions [26,27]. These factors combined to increase the incidence of NMSC in China from 5th in 2018–4th in 2020 [24], a growth rate that exceeded the global average level. The higher mortality and burden of NMSC in China may be related to late diagnosis due to insufficient public awareness of the disease, unequal distribution of healthcare resources, and lack of systematic screening. Although NMSC can usually be cured through surgery, it has been reported that patients with NMSC in China are usually in the middle to late stages when diagnosed, missing the best treatment and having a shorter average survival [28]. Furthermore, China's medical resources are unevenly distributed geographically, with high-quality medical resources more concentrate in large cities and developed coastal areas [29–31]. This may make it difficult for residents in some areas to access timely and effective skin cancer screening and treatment.

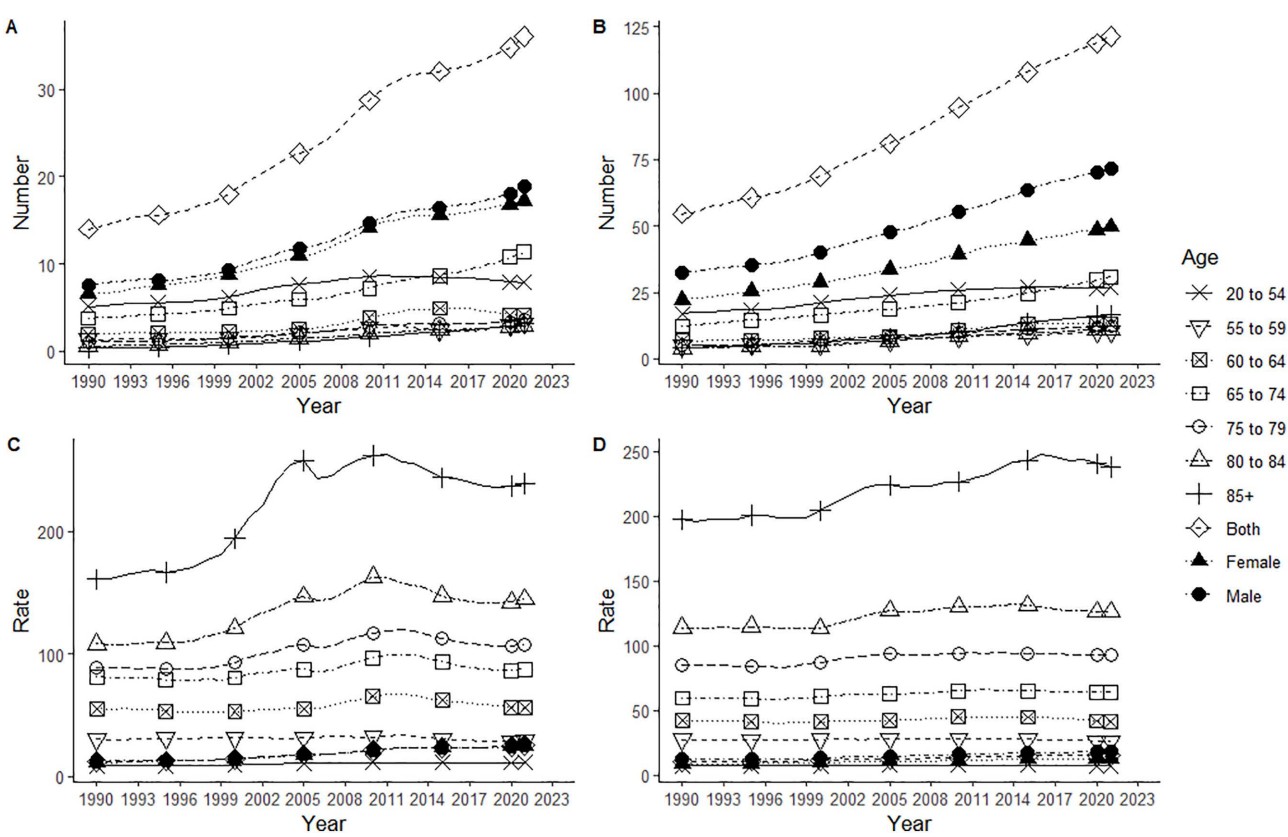

**Fig 5. DALY changes of NMSC in China and global from 1990 to 2021.** DALY in China (A) and Global (B), DALY rate in China (C) and Global (D). DALY, Disability-adjusted life years.

This study further found that the incidence of NMSC was higher in men than in women, both globally and in China. This finding was consistent with existing literature reports, with some studies observing similar patterns of gender differences [32,33]. This difference may be related to the different works and lifestyles of men and women. Men worked outdoors more frequently and were, therefore, more exposed to UV light. In addition, men also used sunscreen, hats, and other protective gear less often than women [19]. At the genetic level, males exhibited higher p53 mutation frequencies than females in various cancers, including esophageal cancer, NMSC, and hepatocellular carcinoma. As a key tumor suppressor gene, p53 plays a central role in DNA repair and cell cycle regulation. Higher p53 mutation rates in male skin cells may increase the risk of developing NMSC, accelerate tumor progression, and result in poorer clinical outcomes. This genetic-level gender difference may provide an important molecular biological explanation for the higher burden of NMSC in men compared to women, and also reflected the foundational influence of gender on cancer susceptibility and development [34].

This study also found that the burden of NMSC tended to increase with age, which was consistent with other studies [14,33]. The possible reason for this could be that human skin is in prolonged contact with the natural environment and exposed to sunlight, thus becoming prone to aging as one gets older [2]. At the same time, the immune system of the elderly is weaker, and their immune function is weakened. Abnormal cells cannot be completely eliminated and gradually proliferate, thereby increasing the risk of cancer. This emphasizes the importance of attention and preventive measures for the elderly [35,36].

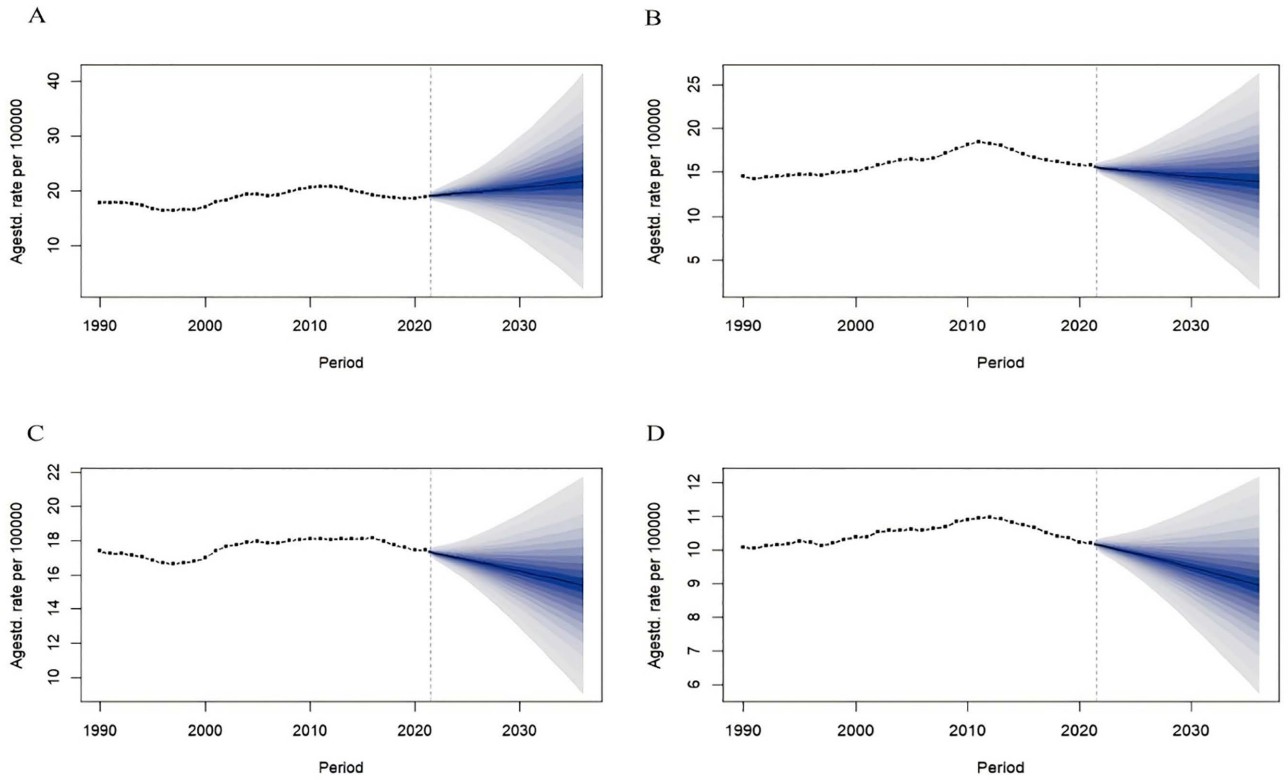

**Fig 6. The ASDR prediction of NMSC in China and global from 2022 to 2036.** ASDR prediction in China male (A), China female (B), Global male (C), Global female (D). Observed (dashed lines) and predicted rates (solid lines). The blue region shows the upper and lower limits of the 95% uncertainty intervals (95% UI). ASDR, Age-standardized disability-adjusted life years rate. NMSC, Non-melanoma skin cancer.

The incidence of NMSC will continue to increase in China and globally from 2022 to 2036, whereas the mortality and DALYs were predicted to decrease gradually. The main reasons for the increase in incidence may be population aging and high-risk behaviors (e.g., increased outdoor recreational activities). China had implemented the "Healthy China Action: Cancer Prevention and Control Implementation Plan (2019-2030)", and the mortality rate of NMSC has been effectively controlled, but there is still a gap compared with European and American countries [37]. Between 2013 and 2017, the introduction of immune checkpoint inhibitors and targeted therapies significantly reduced melanoma mortality rates in the United States [38–40]. Similar therapeutic advancements were gradually applied to NMSC. For example, the PD-1 inhibitor Cemiplimab had been approved by the FDA for advanced cutaneous squamous cell carcinoma, with an objective response rate of up to 47% [41]; the Hedgehog pathway inhibitor Vismodegib achieved an objective response rate of 43% in advanced basal cell carcinoma [42]. Access to these therapies in China is gradually improving, such as the clinical trials of the domestically produced PD-1 inhibitor Toripalimab in skin cancer, and the widespread adoption of early diagnostic technologies (such as dermatoscopy and AI-assisted screening) [43–46], which may drive a reduction in mortality rates and disease burden for NMSC in China. This reflects the progress made in recent years in the early diagnosis and treatment of skin cancer. Effective prevention and treatment of skin cancer play an important role in reducing the incidence and mortality of this disease, especially NMSC. The sustainability of this positive trend needs to be supported by further surveillance and public health interventions.

To address the rising trend in the burden of NMSC, China and other countries implemented a series of targeted policy interventions. Environmental pollution has been proven to impact skin cancer significantly [47]. To this end, China's

Environmental Protection Law, which came into effect on January 1, 2015, provided legal protection for reducing the burden of NMSC by strictly limiting the concentration of environmental pollutants that may increase the risk of NMSC [18]. Through measures such as clean production and pollutant emission control, this law will reduce environmental factors that may increase the risk of NMSC from the source, complementing improvements in healthcare and disease prevention strategies to collectively establish a multi-dimensional policy framework for NMSC prevention and control in China. At the global level, a series of skin cancer prevention and control projects and campaigns focused on reducing the incidence of skin cancer through primary prevention measures such as health education and reducing exposure [48]. Notable initiatives included the "SunPass" project launched by the European Skin Cancer Foundation in 2009 [49] and the "SunSmart" campaign initiated by the UK Cancer Research Center in 2003 [50]. These programs effectively enhanced public awareness of skin cancer prevention through systematic public education, sun protection knowledge dissemination, and intervention in risky behaviors, thereby reducing the risk of skin cancer at its source. As global climate change and environmental pollution persist, NMSC prevention and control will face new challenges. In the future, countries must strengthen policy coordination, integrate environmental protection and public health strategies, and improve monitoring and early warning systems. Additionally, artificial intelligence and large data technologies should be utilized to enhance the accuracy of early cancer screening, innovative technologies and methods should be applied to cancer treatment, and to develop personalized prevention plans and treatment strategies [51–53]. Strengthening international cooperation and sharing prevention and control experiences are crucial for building a global NMSC prevention and control network and reducing the disease burden.

This study had some limitations. The GBD database was a global endeavor to describe the epidemiology of diseases around the world using existing data and sophisticated analytical frameworks but may lack data from vital registries, verbal autopsies, and other sources. In addition, the quantity, quality and calibration methods of the model data also had an impact on the estimated information. Meanwhile, GBD currently lacks data for various regions in China, our study is unable to compare the severity of NMSC in different regions of China and cannot propose regional prevention and control recommendations. This study compared the burden of NMSC in China with that of the world, analyzed differences in age and sex, but did not consider other risk factors, and the study was cross-sectional and could not suggest a causal relationship between relevant risk factors.

## 5. Conclusion

The findings of this study provided valuable information for public health policymakers. In order to address the burden of NMSC, it was recommended that sunscreen education and public awareness of skin cancer risk factors should be strengthened. Future studies should focus on the specific causes of NMSC and how to reduce its burden through effective preventive measures. In addition, studies should consider multiple risk factors, including genetic, environmental, and socioeconomic factors, to gain a more comprehensive understanding of the prevalence of NMSC. In conclusion, this study highlighted the importance of NMSC as a global public health problem and provided a scientific basis for future prevention and intervention measures.

## Acknowledgments

We thank all related works from GBD who devoted their time and energy to preparing these publicly available data.

## Author contributions

**Conceptualization:** Su Liang, Tao Sun, Juanmei Cao, Xue Wang.

**Formal analysis:** Su Liang, Tao Sun, Xuesong Jia, Juanmei Cao, Xue Wang.

**Funding acquisition:** Xuesong Jia, Juanmei Cao, Xue Wang.

**Methodology:** Su Liang, Tao Sun, Xuesong Jia, Juanmei Cao, Xue Wang.

**Project administration:** Xuesong Jia, Juanmei Cao, Xue Wang.

**Software:** Tao Sun, Xuesong Jia, Juanmei Cao, Xue Wang.

**Supervision:** Su Liang, Xuesong Jia, Juanmei Cao, Xue Wang.

**Writing – original draft:** Su Liang, Tao Sun, Juanmei Cao, Xue Wang.

**Writing – review & editing:** Su Liang, Tao Sun, Juanmei Cao, Xue Wang.

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
