## [Decision Letter · Decision Letter 0]

4 Aug 2025

Dear Dr. Cao,

Thank you for submitting your manuscript to PLOS ONE. After careful consideration, we feel that it has merit but does not fully meet PLOS ONE’s publication criteria as it currently stands. Therefore, we invite you to submit a revised version of the manuscript that addresses the points raised during the review process.

We look forward to receiving your revised manuscript.

Kind regards,

Muhammad Ahmad

Academic Editor

PLOS ONE

Journal Requirements:

3. Thank you for stating the following in your manuscript:

“This study was supported by the Research Project of National Natural Science Foundation of China (Project No: 82460624) and Guiding Plan Project of Shihezi Science and Technology Bureau.(Project No: 2024ZDYL07).**”**

4. Please include a separate caption for each figure in your manuscript

Additional Editor Comments (if provided):

Dear Author,

Based on the reviewers’ feedback, the manuscript has received suggestions for both minor and major revisions. My assessment indicates that a major revision is required.

I have reviewed the manuscript and identified several relevant articles that could strengthen your work. I have provided the links to these articles

https://www.pvj.com.pk/pdf-files/25-421.htm

https://www.pvj.com.pk/pdf-files/24-538.pdf

https://www.pvj.com.pk/pdf-files/25-040.pdf

https://doi.org/10.47278/journal.abr/2024.004

https://doi.org/10.47278/journal.abr/2025.008

DOI : 10.9775/kvfd.2024.32444

please review them carefully and cite them appropriately where applicable.

Additionally, all reviewers’ comments must be thoroughly addressed for the manuscript to proceed further in the review process.

Thank you for your attention to these matters.

Reviewers' comments:

Reviewer's Responses to Questions

**Comments to the Author**

1. Is the manuscript technically sound, and do the data support the conclusions?

Reviewer #1: Partly

Reviewer #2: Yes

Reviewer #3: Partly

2. Has the statistical analysis been performed appropriately and rigorously?

Reviewer #1: Yes

Reviewer #2: Yes

Reviewer #3: No

3. Have the authors made all data underlying the findings in their manuscript fully available?

Reviewer #1: Yes

Reviewer #2: Yes

Reviewer #3: Yes

4. Is the manuscript presented in an intelligible fashion and written in standard English?

Reviewer #1: Yes

Reviewer #2: Yes

Reviewer #3: Yes

Reviewer #1: Title: Systematic analysis of non-melanoma skin cancer burden: a comparative study

between China and the world from 1990 to 2021 and prediction to 2036.

I. Overall

1. Abbreviations should be defined at first mention and used consistently thereafter (eg. UV, etc.).

II. Abstract

2. Objective: Kindly enunciate the objective through the use of an action verb.

3. Keywords:

a) Kindly replace "Non-melanoma skin cancer" with "Skin cancer."

b) Although "Bayesian age-period-cohort model" and "Joinpoint regression model" are not directly listed in MeSH, terms like "Bayesian Analysis" and "Regression Analysis" can be suitable alternatives.

c) Kindly replace "Burden of Disease" with "Global Burden of Disease."

III. Introduction

1. Lines 30-32 ► “According to GLOBOCAN […] 8% of all cancer deaths”: This declaration is not substantiated by the cited source. It is important to review the statistics you provided to ensure they are supported by a reference.

2. Lines 35-36 ► “The age-standardized incidence […] to 784 (/100,000) in 2019 [5]”: This declaration is not substantiated by the cited source. You need to adopt a more meticulous approach when presenting statements and statistics.

3. Lines 36-39 ► “Moreover, […] in most countries [6]”: I think you can find a more recent reference.

4. Lines 40-41 ► “At present, […] clinical treatment [7, 8]”: Regarding the main assertion of this passage, I was convinced that the references you mentioned would be systematic reviews or meta-analyses demonstrating that there are very few epidemiological studies on NMSC conducted in China, which is not the case. The two references you wisely cited fail to support this fundamental claim.

5. Lines 42-42 ► “Epidemiological evidence […] interventions”: Isn’t it paradoxical to compare the epidemiological data on NMSC between China and the rest of the world, considering you mentioned in the previous paragraph that there is a lack of data on this type of cancer in China?

6. I believe it would be wise to delve deeper into the epidemiological context of NMSC in China.

IV. Materials and methods

1. To ensure the reproducibility of the data analysis, please indicate whether model non-identifiability was addressed and specify the evaluation criteria.

2. Please provide a reference for the formula you used for the breakpoint regression model.

V. Results

1. Wouldn’t it be wise to isolate age and gender in Figures 1, 3, and 5?

VI. Discussion

1. When the acronym "DALY" is used as a quantifiable measure, it should be employed in the plural form (DALYs).

2. Lines 175-178 ► “Although the level of NMSC […] than that of the world”: This passage must be revised to ensure a balance between precision and fluidity, while also eliminating redundancy and fixing the spelling of the acronym "DLAY."

3. Lines 178-181 ► “There were significant gender […] faster in women”: To enhance the clarity of this text, it would be appropriate to split it into two distinct sentences, highlighting the parallelism between the incidence comparisons and those of mortality/DALYs, while incorporating a smoother transition between the two sentences.

4. Lines 204-205 ► “These factors […] from 5th in 1990 to 4th today”: This sentence does not accurately reflect the cited source [20]. It actually outlined the changes in the most common types of cancer and the main causes of cancer-related mortality during the period from 2018 to 2020, rather than from 1990 to 2025, which is implied by your use of the word "today."

5. Lines 211-213 ► “Second, China’s medical […] developed coastal areas”:

a. The logical connector "Second" that you used is not appropriate;

b. It is imperative to cite one or more references.

6. Lines 216-219 ► “According to […] in men than women [22]”: This passage does not match the indicated reference. The mention of the "American Cancer Society" is inappropriate, as the study was conducted by authors who are individuals. The use of expressions like "twice, three times" to quantify proportions makes it difficult to validate and align with the cited data.

7. Lines 224-228 ► “Firstly, […] of developing cancer”: The use of the past tense in writing this passage deceives the reader; it is phrased in a way that makes it seem like a conclusion, while it actually represents aspects of the discussion.

8. Lines 237-240 ► “Between […] by 6.4% per year”:

a. Firstly, this passage does not match the indicated reference [26];

b. Secondly, if it turns out that this passage is supported by a reference, I believe that is misplaced, particularly as it comes after a discussion on the Chinese context, even though it emphasizes the American context.

VII. References

1. Reference [20] has been incorrectly cited in the final list of references.

2. It is clear that the references need to be verified, especially since the year of publication is often repeated in the entries.

3. It would also be wise to include the DOI when possible.

Reviewer #2: in conclusion sections we need to be more specific and get specific recommendations for specific geographic area , there is need to get a specific location inside China with high pervelance of malenoma

Reviewer #3: General comments:

The study is generally well-structured with time trend and prediction components. However, there are a few important statistical limitations and opportunities for enhancement.

While BAPC is an appropriate choice, the manuscript does not clearly communicate whether posterior distributions, credible intervals, or model diagnostics (e.g., DIC, convergence checks) were evaluated to assess forecast reliability.

The use of national-level aggregates may mask important subnational variations. The authors could consider a Bayesian hierarchical model or spatiotemporal modeling approach (e.g., INLA, STAN) in future work to more precisely estimate and compare disease burden patterns across regions, especially within China. The projections assume the continuation of historical patterns. Incorporating scenario-based forecasts (e.g., demographic shifts, UV index changes, intervention uptake) would make predictions more policy-relevant.

Although this is not a meta-analysis in the strict sense, the paper would benefit from methods used in systematic reviews, such as sensitivity analyses, effect size estimations, and heterogeneity testing, especially when comparing China to global averages.

Detailed comments:

Abstract

A comprehensive grammar and style revision is recommended. The section reads as a rough translation and would benefit from professional language editing or rewriting with better fluency.

“The increase rates (707.31%, 16.00%, and 10.04%) and upward trends -” The wording is unclear and needs to be clarified as to what these percentages refer to. A language and grammar check throughout the abstract is needed to improve readability and academic tone.

The abstract mentions using the Joinpoint regression model and the Bayesian age-period-cohort model. Still, it does not specify how the data were handled, what population demographics were considered, or how the models were validated. A sentence on the scope of data (like population, age range) and model assumptions would improve transparency

In the results, are the percentages the total increase over the period or annual averages? How do the values compare between China and the global average? It is recommended to focus on a few key figures and interpret them succinctly in context.

Introduction

Use the full term first with abbreviation (e.g., "disability-adjusted life years (DALYs)") and remain consistent in using “DALYs” instead of alternating between singular and plural forms.

Line 32: You mention "8% of all cancer deaths," but this conflicts with the relatively low mortality usually associated with NMSC. Double-check this figure.

Line 40: The statement “relatively few epidemiological studies on NMSC in China” is vague. You should briefly cite existing national-level epidemiological sources (if any) to support the claim, or clarify whether the lack is due to underreporting, registry limitations, or low research focus.

Materials and Methods

What ICD codes were used to define NMSC in the GBD dataset? (e.g., ICD-10 C44 for NMSC?) Were age-standardized rates obtained directly from GBD or calculated independently? Clarify the age-standardization process (e.g., using which standard population). What age groups were included in the study? How were male and female data handled? Were trends analyzed separately by sex for each metric? Cite the methods chosen by the previous literature and mention/quote your selection criteria for a specific test.

The model should be described more formally: Did you use the National Cancer Institute’s Joinpoint Regression Program? Specify the full name and source for software: “Joinpoint Regression Program (version 5.0.2) developed by the U.S. National Cancer Institute.”

What significance level was used to identify joinpoints? Were there any constraints on the number of joinpoints? Was model selection based on a permutation test?

In the Bayesian age-period cohort model, what R package was used? What were the prior assumptions, time intervals, or model parameters? How was model fit assessed?

This section is incomplete without including BAPC modeling details, which are critical for prediction validity.

Results

Lines 99-103: The last part about age groups is confusing and partially contradictory. You say: “the lowest was 85+ years old,” then shortly after, say “highest in the world was 85+ years.” Also, 20–54 years is described as the lowest age-specific incidence, but that contradicts earlier groupings. Clarify this.

Lines 141-148: While the data is informative, there’s little attempt to interpret or explain what these DALY trends might mean (e.g., differences in disability impact, treatment access, demographic effects).

Discussion

The discussion blends a summary of results, external context, and policy implications without clear thematic organization. It begins with a generic overview and then jumps to specific issues like tanning, ozone depletion, and tourism without adequate linkage. There’s a lack of focused structure.

The key finding that China has a lower incidence but faster increases and relatively high DALY/mortality burdens deserves deeper analysis. Is this due to late diagnosis, limited access to dermatologic care, or underreporting of non-lethal cases? Why are DALYs and deaths rising faster than globally, despite lower baseline rates?

Statements like “this may be related to…” or “it has been shown that…” are too vague. Citations [14] to [17] are not explained or integrated. It is unclear what study [14] found and how it aligns or differs from your own results. Clearly describe each cited study and explicitly link it to your findings.

Consider gender-specific behaviors (e.g., tanning practices, occupational exposures, cosmetic trends), and whether care-seeking or diagnosis patterns differ. Expand discussion beyond UV exposure. Consider biological factors (e.g., hormone-related differences in immune response), or differences in early detection and healthcare access. Consider referencing recent studies on sex-specific tumor behavior, if available, or national screening trends. Use established concepts like cumulative UV exposure, immunosenescence, or actinic damage to describe the mechanisms more accurately. Please see Sarwar Z., Nomi Z.A., Awais M., Shahbakht R.M., et al. (2025) for a broader discussion on the impact of climate change on disease patterns, including environmental triggers such as increased UV exposure that may contribute to the rising burden of skin-related diseases. Effect of Climate Change on Transmission of Livestock Diseases. Agrobiological Records, 19, 1–11. https://doi.org/10.47278/journal.abr/2025.001

Include mention of improved early detection, increased awareness, and treatment advances (as you briefly touch on with the melanoma example). Clarify whether such progress applies to NMSC specifically. The example about melanoma treatments (e.g., checkpoint inhibitors) may not apply unless explicitly linked. For integrating plant-based compounds into zoonotic disease control strategies, particularly for their potential anthelmintic and immunomodulatory effects, please see Alsayeqh, A.F. (2025). Botanicals: A Promising Control Strategy Against Highly Zoonotic Foodborne Trichinosis. Kafkas Üniversitesi Veteriner Fakültesi Dergisi, 31(2), 129–136. https://doi.org/10.9775/kvfd.2023.32154

Include one or two concrete examples of relevant NMSC-related actions under the policy, or state the lack thereof. The section contains meaningful points but needs a more precise, evidence-backed explanation of observed trends, especially regarding aging, gender, and future burden. It should also clarify distinctions between NMSC and melanoma, and better integrate policy and prevention contexts.

**Do you want your identity to be public for this peer review?** For information about this choice, including consent withdrawal, please see our Privacy Policy

Reviewer #1: **Yes: ** Hicham EL MOUADDIB, PhD

Reviewer #2: No

Reviewer #3: No

---

## [Author Response · Author response to Decision Letter 1]

27 Aug 2025

PLOS ONE

Aug 28, 2025

PONE-D-25-28458

Systematic analysis of non-melanoma skin cancer burden: a comparative study between China and the world from 1990 to 2021 and prediction to 2036

Dear editor and reviewers,

We appreciate editor and reviewers very much for their positive and constructive comments and suggestions on our manuscript entitled “Systematic analysis of non-melanoma skin cancer burden: a comparative study between China and the world from 1990 to 2021 and prediction to 2036.” (Manuscript ID: PONE-D-25-28458). Those comments are all valuable and very helpful for revising and improving our paper, as well as the important guiding significance to our researches. Based on these comments and suggestions, we have made careful modification on the original manuscript.

On the separate pages, we provided our response to the comments and suggestions, point by point, and highlighted the changes in the marked copy of the revision. We hope that our revision will be approved by the experts and reviewed favorably.

Sincerely,

Juan Mei Cao, MD

Reviewer #1: Title: Systematic analysis of non-melanoma skin cancer burden: a comparative study between China and the world from 1990 to 2021 and prediction to 2036.

I. Overall

1. Abbreviations should be defined at first mention and used consistently thereafter (eg. UV, etc.).

Response Thank you very much for your careful review. We have revised the full text.

II. Abstract

1. Objective: Kindly enunciate the objective through the use of an action verb.

Response Thank you for your reasonable suggestion. We have made the revisions in lines 4-6: “To compare the characteristics and trends of the non-melanoma skin cancer (NMSC) burden in China and globally, and to provide a basis for the development of effective prevention and control measures in China.”

2. Keywords:

a) Kindly replace "Non-melanoma skin cancer" with "Skin cancer."

b) Although "Bayesian age-period-cohort model" and "Joinpoint regression model" are not directly listed in MeSH, terms like "Bayesian Analysis" and "Regression Analysis" can be suitable alternatives.

c) Kindly replace "Burden of Disease" with "Global Burden of Disease."

Response Thank you for your professional advice. We have modified the key words in lines 29-30: “Skin cancer; Regression analysis; Bayesian analysis; Global burden of disease; Comparative study.”

III. Introduction

1. Lines 30-32 ► “According to GLOBOCAN […] 8% of all cancer deaths”: This declaration is not substantiated by the cited source. It is important to review the statistics you provided to ensure they are supported by a reference.

Response: Thank you very much for your professional advice. We are extremely sorry for this error and have checked and re-described it in lines 34-36: “According to GLOBOCAN, there were 1,234,595 new cases and 69,481 deaths of NMSC globally in 2022, accounting for 0.7% of all cancer deaths[2].”

2. Lines 35-36 ► “The age-standardized incidence […] to 784 (/100,000) in 2019 [5]”: This declaration is not substantiated by the cited source. You need to adopt a more meticulous approach when presenting statements and statistics.

Response: Thank you for your careful review. We have made the necessary corrections in lines 39-41: “The age-standardized incidence rate (ASIR) of NMSC in the United States had increased from 402 (/100,000) in 1990 to 787 (/100,000) in 2019[5].”

3. Lines 36-39 ► “Moreover, […] in most countries [6]”: I think you can find a more recent reference.

Response Thank you for your reasonable suggestions. We have reviewed and added to, as well as updated, the relevant supporting literature in lines 41-49: “However, the actual incidence of NMSC may be underestimated, potentially due to the following reasons: many countries did not require reporting NMSC information to national cancer registries, which directly led to most countries not giving sufficient attention to NMSC[6-9]; the high incidence of NMSC not only resulted in a large workload and difficulty in implementing comprehensive registration, but also posed technical challenges in accurately recording each patient's tumor status and extracting structured pathological data, making it difficult to obtain relevant data[10]. These factors collectively contribute to the fact that the actual burden of NMSC remains unclear and is frequently underestimated[11].”

4. Lines 40-41 ► “At present, […] clinical treatment [7, 8]”: Regarding the main assertion of this passage, I was convinced that the references you mentioned would be systematic reviews or meta-analyses demonstrating that there are very few epidemiological studies on NMSC conducted in China, which is not the case. The two references you wisely cited fail to support this fundamental claim.

Response: We fully agree with your suggestion. We made the necessary revisions in lines 50-52: “Compared with melanoma, NMSC has a lower mortality rate and better prognosis. This difference has led to limited scientific evidence on the epidemiological characteristics and burden of NMSC in China[12].”

5. Lines 42-42 ► “Epidemiological evidence […] interventions”: Isn’t it paradoxical to compare the epidemiological data on NMSC between China and the rest of the world, considering you mentioned in the previous paragraph that there is a lack of data on this type of cancer in China?

Response: We fully agree with your point of view. We made revisions in lines 53-60 to make the logic more reasonable: “Therefore, in order to better address the challenges posed by NMSC, this study, based on the Global Burden of Disease (GBD) 2021 database, used the Joinpoint regression model and the Bayesian age-period-cohort (BAPC) model first to analyze and predict the burden (incidence rate, mortality rate, and disability-adjusted life years (DALYs)) caused by NMSC in China from 1990 to 2021 by gender, and then compared it with the global situation, to provide a basis for formulating more effective prevention and treatment strategies and intervention measures.”

6. I believe it would be wise to delve deeper into the epidemiological context of NMSC in China.

Response: We fully agree with your opinion and accept it entirely. We have re-examined the relevant literature and revised the introduction section. We are very much looking forward to your approval.

IV. Materials and methods

1. To ensure the reproducibility of the data analysis, please indicate whether model non-identifiability was addressed and specify the evaluation criteria.

Response: Thank you for your scientific suggestions. We use the coefficient of determination R2 to evaluate the predictive performance of the BAPC model. The larger the R2 value, the better the model fits the data. We made corresponding modifications in lines 110-113: “To verify the accuracy of the model, we calculated the coefficient of determination (R²) for each BAPC model. R² reflects the extent to which the model explains the variation in the data. The closer the value is to 1, the better the model fit and the higher the prediction accuracy.”

In lines 145-147: We used the BAPC model to predict the ASIR of NMSC in China and globally based on different genders. The model fit was excellent (R2China male=0.99998, R2China female=0.99998; R2Global male=0.96204, R2Global female=0.98055).

In lines 177-179: We used the BAPC model to predict the ASMR of NMSC in China and globally based on different genders. The model fit was excellent (R2China male=0.99643, R2China female=0.98751; R2Global male=0.86609, R2Global female=0.98699).

In lines 210-212: We used the BAPC model to predict the ASDR of NMSC in China and globally based on different genders. The model fit was excellent (R2China male=0.99987, R2China female=0.99949; R2Global male=0.97468, R2Global female=0.99812).

2. Please provide a reference for the formula you used for the breakpoint regression model.

Response: Thank you very much for your suggestion. We have added the references in lines 99:“The model formula was as follows[14]”

V. Results

1. Wouldn’t it be wise to isolate age and gender in Figures 1, 3, and 5?

Response Thank you very much for your suggestion. These three figures are mainly designed to show the trend over time. We believe that combining age and gender can help observe the trend and at the same time avoid presenting more similar figures. We hope you can understand this.

VI. Discussion

1. When the acronym "DALY" is used as a quantifiable measure, it should be employed in the plural form (DALYs).

Response: Thank you very much for your careful review. We have made revisions to the entire text.

2. Lines 175-178 ► “Although the level of NMSC […] than that of the world”: This passage must be revised to ensure a balance between precision and fluidity, while also eliminating redundancy and fixing the spelling of the acronym "DLAY."

Response: We are grateful for your reasonable suggestions. We have made the necessary changes in lines 225-228: “Although the ASIR of NMSC in China was lower than the global level, the levels of both ASMR and ASDR were relatively higher. Meanwhile, the increased rates of the three indicators for NMSC in China were faster than that in the world.”

3. Lines 178-181 ► “There were significant gender […] faster in women”: To enhance the clarity of this text, it would be appropriate to split it into two distinct sentences, highlighting the parallelism between the incidence comparisons and those of mortality/DALYs, while incorporating a smoother transition between the two sentences.

Response: Thank you very much for your patient review. We have made the necessary changes in lines 228-231: “There were significant gender differences in the burden of NMSC: the incidence and its increase rate among men were both higher than those of women; the mortality and DALYs rate for men were more severe, but their increase rates for women were faster.”

4. Lines 204-205 ► “These factors […] from 5th in 1990 to 4th today”: This sentence does not accurately reflect the cited source [20]. It actually outlined the changes in the most common types of cancer and the main causes of cancer-related mortality during the period from 2018 to 2020, rather than from 1990 to 2025, which is implied by your use of the word "today."

Response: Thank you very much for your careful review. We have made the necessary corrections in lines 287-288: “These factors combined to increase the incidence of NMSC in China from 5th in 2018 to 4th in 2020[24], a growth rate that exceeded the global average level.”

5. Lines 211-213 ► “Second, China’s medical […] developed coastal areas”:

a. The logical connector "Second" that you used is not appropriate;

b. It is imperative to cite one or more references.

Response: We fully agree with your point of view. We have made revisions and additions in lines 291-296: “Although NMSC can usually be cured through surgery, it has been reported that patients with NMSC in China are usually in the middle to late stages when diagnosed, missing the best treatment and having a shorter average survival[28]. Furthermore, China's medical resources are unevenly distributed geographically, with high-quality medical resources more concentrate in large cities and developed coastal areas[29–31].”

6. Lines 216-219 ► “According to […] in men than women [22]”: This passage does not match the indicated reference. The mention of the "American Cancer Society" is inappropriate, as the study was conducted by authors who are individuals. The use of expressions like "twice, three times" to quantify proportions makes it difficult to validate and align with the cited data.

Response: Thank you very much for bringing this issue to our attention. We have made revisions and additions in lines 299-313 of the revised version: “This study further found that the incidence of NMSC was higher in men than in women, both globally and in China. This finding was consistent with existing literature reports, with some studies observing similar patterns of gender differences[32,33]. This difference may be related to the different works and lifestyles of men and women. Men worked outdoors more frequently and were, therefore, more exposed to UV light. In addition, men also used sunscreen, hats, and other protective gear less often than women[19]. At the genetic level, males exhibited higher p53 mutation frequencies than females in various cancers, including esophageal cancer, NMSC, and hepatocellular carcinoma. As a key tumor suppressor gene, p53 plays a central role in DNA repair and cell cycle regulation. Higher p53 mutation rates in male skin cells may increase the risk of developing NMSC, accelerate tumor progression, and result in poorer clinical outcomes. This genetic-level gender difference may provide an important molecular biological explanation for the higher burden of NMSC in men compared to women, and also reflected the foundational influence of gender on cancer susceptibility and development[34].”

7. Lines 224-228 ► “Firstly, […] of developing cancer”: The use of the past tense in writing this passage deceives the reader; it is phrased in a way that makes it seem like a conclusion, while it actually represents aspects of the discussion.

Response: We completely agree with you. We made the modifications in lines 314-321: “The possible reason for this could be that human skin is in prolonged contact with the natural environment and exposed to sunlight, thus becoming prone to aging as one gets older[2]. At the same time, the immune system of the elderly is weaker, and their immune function is weakened. Abnormal cells cannot be completely eliminated and gradually proliferate, thereby increasing the risk of cancer. This emphasizes the importance of attention and preventive measures for the elderly[35,36].”

8. Lines 237-240 ► “Between […] by 6.4% per year”:

a. Firstly, this passage does not match the indicated reference [26];

b. Secondly, if it turns out that this passage is supported by a reference, I believe that is misplaced, particularly as it comes after a discussion on the Chinese context, even though it emphasizes the American context.

Response: Thank you very much for bringing this issue to our attention. We have made revisions and additions in lines 328-339: “Between 2013 and 2017, the introduction of immune checkpoint inhibitors and targeted therapies significantly reduced melanoma mortality rates in the United States[38–40]. Similar therapeutic advancements were gradually applied to NMSC. For example, the PD-1 inhibitor Cemiplimab had been approved by the FDA for advanced cutaneous squamous cell carcinoma, with an objective response rate of up to 47%[41]; the Hedgehog pathway inhibitor Vismodegib achieved an objective response rate of 43% in advanced basal cell carcinoma[42]. Access to these therapies in China is gradually improving, such as the clinical trials of the domestically produced PD-1 inhibitor Toripalimab in skin cancer, and the widespread adoption of early diagnostic technologies (such as dermatoscopy and AI-assisted screening)[43–46], which may drive a reduction in mortality rates and disease burden for NMSC in China.”

VII. References

1. Reference [20] has been incorrectly cited in the final list of references.

Response: Thank you very much for your question. Modified.

2. It is clear that the references need to be verified, especially since the year of publication is often repeated in the entries.

Response: Thank you very much for your question. Modified.

3. It would also be wise to include the DOI when possible.

Response: Thank you very much for your question. We have added.

Reviewer #2: in conclusion sections we need to be more specific and get specific recommendations for specific geographic area , there is need to get a specific location inside China with high pervelance of malenoma

Response Thank you very much for your professional advice. Our study was conducted based on the GBD database. However, GBD currently does not provide data for various regions in China, so we are currently unable to analyze the burden of NMSC in different regions of China and cannot determine which region is more severely affected. This is the limitation of this study. We have added a description of the limitations in the discussion section, hoping for your understanding, lines 378-381: “Meanwhile, GBD cur

---

## [Decision Letter · Decision Letter 1]

21 Sep 2025

Systematic analysis of non-melanoma skin cancer burden: a comparative study between China and the world from 1990 to 2021 and prediction to 2036

PONE-D-25-28458R1

Dear Dr. Cao,

We’re pleased to inform you that your manuscript has been judged scientifically suitable for publication and will be formally accepted for publication once it meets all outstanding technical requirements.

Kind regards,

Muhammad Ahmad

Academic Editor

PLOS ONE

Additional Editor Comments (optional):

Reviewer #1:

Reviewer #2:

Reviewers' comments:

Reviewer's Responses to Questions

**Comments to the Author**

Reviewer #1: All comments have been addressed

Reviewer #2: All comments have been addressed

2. Is the manuscript technically sound, and do the data support the conclusions?

Reviewer #1: Yes

Reviewer #2: Yes

3. Has the statistical analysis been performed appropriately and rigorously?

Reviewer #1: Yes

Reviewer #2: Yes

4. Have the authors made all data underlying the findings in their manuscript fully available?

Reviewer #1: Yes

Reviewer #2: Yes

5. Is the manuscript presented in an intelligible fashion and written in standard English?

Reviewer #1: Yes

Reviewer #2: Yes

Reviewer #1: Dear authors,

Thank you for the effort you put into improving the manuscript for this revision.

At the end of this second review, I retained only two remarks. The first concerns the first formula you provided. Apparently, due to a font or formatting issue (Line 80), this was not properly edited. Moreover, you should provide a reference for that. The second remark concerns reference [8], for which you should provide an internet link.

Regards,

Reviewer #2: (No Response)

**Do you want your identity to be public for this peer review?** For information about this choice, including consent withdrawal, please see our Privacy Policy

Reviewer #1: No

Reviewer #2: No

---

## [Editor Report · Acceptance letter]

PONE-D-25-28458R1

PLOS ONE

Dear Dr. Cao,

I'm pleased to inform you that your manuscript has been deemed suitable for publication in PLOS ONE. Congratulations! Your manuscript is now being handed over to our production team.

Kind regards,

on behalf of

Mr. Muhammad Ahmad

Academic Editor

PLOS ONE